# Human IPSC-Derived Model to Study Myelin Disruption

**DOI:** 10.3390/ijms22179473

**Published:** 2021-08-31

**Authors:** Megan Chesnut, Hélène Paschoud, Cendrine Repond, Lena Smirnova, Thomas Hartung, Marie-Gabrielle Zurich, Helena T. Hogberg, David Pamies

**Affiliations:** 1Center for Alternatives to Animal Testing (CAAT), Johns Hopkins Bloomberg School of Public Health, 615 N Wolfe St., Baltimore, MD 21205, USA; meganechesnut@gmail.com (M.C.); lena.smirnova@jhu.edu (L.S.); thartun1@jhu.edu (T.H.); 2Department of Biomedical Sciences, University of Lausanne, CH-1005 Lausanne, Switzerland; helene.paschoud@hotmail.com (H.P.); cendrine.repond@unil.ch (C.R.); marie-gabrielle.zurichfontanellaz@unil.ch (M.-G.Z.); 3Center for Alternative to Animla Testing Europe, University of Konstanz, 78464 Konstanz, Germany; 4Swiss Centre for Applied Human Toxicology (SCAHT), 4055 Basel, Switzerland

**Keywords:** developmental neurotoxicity, neurotoxicity, organotypic, organoid, myelin, developmental diseases, oligodendrocytes

## Abstract

Myelin is of vital importance to the central nervous system and its disruption is related to a large number of both neurodevelopmental and neurodegenerative diseases. The differences observed between human and rodent oligodendrocytes make animals inadequate for modeling these diseases. Although developing human in vitro models for oligodendrocytes and myelinated axons has been a great challenge, 3D cell cultures derived from iPSC are now available and able to partially reproduce the myelination process. We have previously developed a human iPSC-derived 3D brain organoid model (also called BrainSpheres) that contains a high percentage of myelinated axons and is highly reproducible. Here, we have further refined this technology by applying multiple readouts to study myelination disruption. Myelin was assessed by quantifying immunostaining/confocal microscopy of co-localized myelin basic protein (MBP) with neurofilament proteins as well as proteolipid protein 1 (PLP1). Levels of PLP1 were also assessed by Western blot. We identified compounds capable of inducing developmental neurotoxicity by disrupting myelin in a systematic review to evaluate the relevance of our BrainSphere model for the study of the myelination/demyelination processes. Results demonstrated that the positive reference compound (cuprizone) and two of the three potential myelin disruptors tested (Bisphenol A, Tris(1,3-dichloro-2-propyl) phosphate, but not methyl mercury) decreased myelination, while ibuprofen (negative control) had no effect. Here, we define a methodology that allows quantification of myelin disruption and provides reference compounds for chemical-induced myelin disruption.

## 1. Introduction

Myelination is the process in which myelin sheaths are formed by oligodendrocytes in the central nervous system (CNS) and Schwann cells in the peripheral nervous system (PNS). Myelin sheaths consist of a vast variety of lipids and proteins [1]. Oligodendrocytes, by forming a plasma membrane extension that wraps neuronal axonal segments, generate this structure necessary for the saltatory impulse propagation [2,3]. As half of the human brain is composed of myelinated axons (white matter), damage or loss of this structure or to the producing cells could be detrimental to the central nervous system. Many neurodegenerative diseases and neurological disorders are associated with impaired myelin integrity, including multiple sclerosis, acute-disseminated encephalomyelitis, acute hemorrhagic leucoencephalitis, Alzheimer’s, Parkinson’s, and Huntington’s diseases [4,5,6,7] and more recently Williams syndrome [4,5,6,7,8,9]. Myelin deficits have shown social impairments, motor abnormalities and poor white matter functioning [9]. The study of myelin has mainly been conducted in animals, due to the difficulty of modeling myelination in vitro [10]. However, there are clear differences between rodents and humans [11]. Differential gene expression analysis revealed 244 genes expressed in human oligodendrocytes that are not expressed in their mouse counterparts [12]. In addition, the protein composition of myelin is not fully conserved across species [13]. Thus, there is a need to develop models that better represent human myelin physiology.

The brain is the most complex organ in the body in terms of development and function, making it extremely vulnerable to insults [14]. Small changes during development may lead to severe neurological problems. Moreover, infants and children have higher air, food, and water intake per body weight [15,16], immature metabolism [17,18], and certain behaviors such as hand-to-mouth, crawling, and playing close to the ground [16,17], making them more susceptible to chemically-induced brain damage than adults [15,16]. Perturbations to proliferation, migration, cell composition, neuronal network formation, and cell organization, for example, can have detrimental consequences for the function of the brain [18,19]. Most of the time, these effects are not detectable shortly after birth but manifest later in life [20].

Studies between 1997 and 2008 showed a 17% increase in the prevalence of developmental disabilities in children in that decade compared to the previous [21]. Centers for Disease Control and Prevention (CDC), for example, estimated that one in 54 children born in the US was diagnosed with autism spectrum disorder (ASD) [22], and reported that 9.4% (6.1 million) children in United States were diagnosed with attention deficit/hyperactivity disorder (ADHD) in 2016 [23]. In addition to improved diagnostics, medical advances, and improved survival rates of (pre-term) newborns [21,24], environmental contaminants may play a substantial role [18,20,25] in this increase. Despite enhanced research in the last years, the majority of etiologies of these neurodevelopmental disorders are still unknown [20,26], while concerns about the influence of environmental contributions are rising [27,28,29]. Furthermore, some evidence indicates that environmental chemicals, e.g., lead, arsenic, and methylmercury, can contribute to subclinical neurodevelopmental toxicity (such as decreased cognitive function) [20,30,31].

The extensive costs, time, and difficulty to interpret data from current in vivo testing guidelines for developmental neurotoxicity (DNT), makes it one of the least tested toxicological hazards [32,33]. Therefore, the lack of information for most of the chemicals present on the market, including the high-volume production chemicals [34,35], becomes a very important public health concern. The use of in vitro methods present a more cost-effective and potentially more human-relevant alternative for DNT [35]—moving from observational animal experimentation to a mechanism-based science [36,37,38,39,40]. A comprehensive in vitro DNT strategy consisting of a testing battery of assays that recapitulate key processes during brain development, such as migration, proliferation, differentiation, synaptogenesis, myelination, apoptosis, and neuronal network formation, has been suggested [19,41,42,43]. However, assessment of the myelination process is challenging due to the lack of suitable in vitro models that can recapitulate the in vivo situation. Various models that evaluate oligodendrocyte toxicity can be found in the literature and have been reviewed elsewhere [10]. These models are primarily monocultures of oligodendrocyte precursors or oligodendrocytes, deprived of the cell-cell interactions with neurons required to achieve myelination. Although co-culturing of neurons and oligodendrocytes exist [44,45,46], most of these cultures are still in a monolayer condition, which difficult myelination. However, some co-cultures have been able to generate compact myelin over neuronal axons [47,48] even using stem cell-derived human [49]. Another in vitro models exhibit intense myelination, such as ex vivo rodent brain slices [50,51,52], rat aggregating brain cell cultures [53,54,55,56], or microfluidics cell culture systems [57], but they are all animal-based. Since myelin composition is not well conserved across species [13], DNT testing absolutely requires human cell-based models. Human IPSC-derived models to study myelin disruption have been revised elsewhere [58]. Some new human models combining iPSC and novel in vitro cultures technologies [59,60,61,62,63] have emerged, but only two study showed quantitative assessment [63]. Here, we provide an extensive evaluation of myelin quantification and adequate reference compounds for the assessment of myelin disruption.

The organotypic human BrainSpheres model, developed at Johns Hopkins University [61,64], mimics several steps of brain development [61]. The model, derived from iPSCs, is reproducible in terms of size, shape, and cell composition; contains most major brain cell types, such as different types of neurons, oligodendrocytes, and astrocytes [61]; and presents spontaneous neuronal electrical activity, with around 40% of myelinated axons [61]. Previous results have shown its usefulness for studying pharmacotoxicology [65], brain diseases [66], neurotoxicology [67], and DNT [68,69]. For the latter, the model has shown to adequately assess deleterious chemical effects on key processes of brain development, such as neurite outgrowth, synaptogenesis, and oligodendrocyte differentiation [68,69].

The aim of this study was to evaluate the potential of the BrainSphere model to serve as a myelination test system and to evaluate a BrainSphere myelination assay (Figure 1A). Since only a limited number of reference compounds perturbing myelination have been identified as DNT toxicants, an extensive systematic literature review to identify positive and negative test chemicals for assay development was performed.

## 2. Results

### 2.1. Selection of Chemicals for Assay Development

#### 2.1.1. Literature Review Results

The literature review search (Table 1) yielded 5223 results in PubMed as of 15 November 2017. Once the PMID list was imported into SWIFT-Review, the program organized articles were sorted in categories based on tags automatically assigned according to pre-defined search filters for chemicals and topics prepared by information scientists at NIEHS and EPA, as well as on data imported with articles from PubMed, including MeSH terms, supplementary concept records, keywords, publication type, and Medline indexing tags. The majority of studies retrieved in this literature review were tagged as focusing on developmental and neurological outcomes (Figure 1D) and general environmental exposures (Figure 1E).

Once 5% of articles (262) had been screened based on inclusion and exclusion criteria (Table 2), studies were automatically prioritized in SWIFT-Review, and as expected, more than 90% of the training set of articles placed within the top 10% of the ranked list. For the test set of articles, approximately 85% placed within the top 10% of the ranked list and nearly all of the test set occurred within the top 25% of the ranked list (Figure 1F). Given these results, only studies ranked highly (>0.7; 187 articles) were chosen for further review. Abstract review of these studies was conducted according to inclusion and exclusion criteria (Table 2) and this resulted in the inclusion of 143 studies for full text evaluation. Following a full text review, a list of potential test chemicals was generated, containing nine substances (Appendix A). This list included cuprizone [70], toluene [71], bisphenol A (BPA) [72,73], BDE-99 [74,75], ethanol [76,77,78,79], methyl mercury [80], Tris(1,3-dichloro-2-propyl) phosphate (TDCPP) [81], vanadium [82,83,84], and lead [85,86] (Figure 1G). For this study, only four chemicals were selected from this list.

#### 2.1.2. Selection of Test Chemicals from Literature Review

We selected cuprizone, a copper chelator, as a positive reference compound for myelin assay development for its known ability to induce demyelination and oligodendrocyte-specific cell death if exposure continued [87]. Cuprizone is extensively used as a standard demyelinating agent in animal studies of multiple sclerosis [88,89]. In addition, three environmental chemicals, potentially able to disrupt myelin, were selected from the literature review. Methyl mercury is a well-known DNT compound and three studies found in the literature review identified it as a disruptor of myelination during neurodevelopment. Furthermore, perinatal exposure to methyl mercury was found to reduce the oligodendrocyte cell population and alter MBP gene expression in the developing rat cerebellum [80]. Another selected chemical—one of the most common flame retardants, TDCPP—is progressively replacing polybrominated diphenyl ethers (PBDEs). Recently, high concentrations of TDCPP were detected in polyurethane foam in furniture and house dust in the U.S. at levels that matched previous levels of PBDEs [90]. TDCPP has also recently been found to be neurotoxic and to alter neural differentiation in vitro [91]. In addition, it has been demonstrated to modify MBP mRNA and protein expression in zebrafish larvae, possibly acting through thyroid hormone disruption [81]. The third environmental compound, BPA, is an endocrine disruptor with estrogenic activity that is found in many consumer products. It has been shown to impair myelination in rat brain hippocampus development [72] as well as to reduce the population of oligodendrocytes in the rat hippocampus [73].

Finally, ibuprofen, a non-steroidal anti-inflammatory drug (NSAID), was selected as a negative reference compound for assay development, based on the recommendations given in a recent review on reference compounds for alternative DNT test methods [92]. This is a particularly relevant compound for our study as it crosses the BBB [93], has not been demonstrated to elicit DNT, and is a drug accepted during pregnancy and for children [92]. Ibuprofen has also been used as a negative reference compound in a recent in vitro toxicity study on neural cells derived from human embryonic stem cells [94].

This review represents a first step in the effort to identify reference compounds to study perturbation of the myelination process in vitro during brain development. The chemicals for this study were selected to represent various chemical classes and mechanisms of action to demonstrate the robustness and versatility of the BrainSphere model for DNT testing.

### 2.2. Maturation of Oligodendrocyte and Expression of Myelin-Related Markers during BrainSphere Differentiation

To characterize glial maturation in the BrainSpheres, expression of oligodendrocyte-, myelin-, and astrocyte-specific genes was assessed at two, four, and eight weeks of differentiation and compared to the expression found in NPCs. *NG2* gene expression, a marker of oligodendrocyte precursors [95], was significantly decreased in BrainSpheres compared to NPCs (Figure 2B). The mRNA levels of both oligodendrocyte progenitor markers *OLIG1* and *OLIG2* were significantly increased, as BrainSpheres differentiated (Figure 2B), whereas *CNP* expression did not show any significant changes. *SOX10* expression slightly decreased at two and four weeks of differentiation and further increased at eight weeks, but this upregulation was not statistically significant (Figure 2B). The mRNA levels of the myelin-specific markers *PLP1*, *MBP* and *MOG* increased with time in culture, but this was significant only for *PLP1* (Figure 2B). Altogether, these gene expression patterns correspond to oligodendrocyte maturation. The results of qRT-PCR were confirmed by immunostainings. The oligodendrocyte marker sulfatide O4 and myelin markers MBP and PLP1 were evaluated in BrainSphere differentiation at two, four, and eight weeks. MBP presented a very low expression at two weeks, being stronger at four weeks and eight weeks (Figure 2C). The quantification of MBP pixels showed a strong increase between Week 4 and Week 8 (Figure 2D), indicating maturation of myelin starts after four weeks in culture. PLP1, on the contrary, was only expressed at the eight-week time point (Figure 2G).

In the case of oligodendrocytes, it was observed that the number of O4-positive cells increases from two to four weeks, showing higher maturation at eight weeks of differentiation (Figure 2F). The staining at eight weeks is also more notable, and O4 positive cells also appear to have more processes, which is characteristic of mature oligodendrocytes (Figure 2F). These results suggest the presence of fewer but more mature oligodendrocytes, indicating a continuing maturation process.

Furthermore, the continuing maturation of neurons is observed by the increasing number of pixels of the NF200 immunostaining with time in culture (Figure 2C–E), and for astrocytes by the increase in GFAP and S100β immunofluorescence (Figure 2F,G).

### 2.3. Exposure to DNT Selected Compounds Alter Myelin in BrainSpheres

#### 2.3.1. Cytotoxicity Assessment of Test Chemicals

BrainSpheres were exposed to a range of concentrations (0.1–100 μM) of the test compounds for seven days. Samples were collected on Day 7 of exposure and cytotoxicity was determined using the resazurin assay. Cuprizone, BPA, and ibuprofen did not induce any change in BrainSpheres viability at the tested concentrations (Figure 2H). Methylmercury chloride was found cytotoxic at 10 during the latest stage of differentiation (Figure 2H). At 4/4 none of the concentrations showed statistically significant cytotoxicity (data not shown).

#### 2.3.2. Myelin Is Affected after Exposure of BrainSpheres to Chemicals Inducing DNT

MBP expression was quantified in confocal images of BrainSpheres after exposure to the test chemicals following the treatment schemes 4/8 and 8/8 (Figure 1C). No changes in myelin were observed in 4/8 exposure to any of the compounds (data not shown). Briefly, confocal images were converted in binary images (Figure 3A) and then NF200 (axonal) pixels and MBP (myelin) pixels were overlapped and quantified using Kerman protocol [57]. A significant decrease in the percentage of BrainSphere myelination was observed when exposed to all test chemicals during the eighth week of differentiation (8/8), except for ibuprofen (Figure 3B,C). A concentration-dependent decrease in myelination was observed after exposure to positive reference compound cuprizone (Figure 3C). Exposure to methyl mercury chloride resulted in a significant decrease in myelination only at 10 μM (Figure 3C), which was a cytotoxic concentration (8/8; Figure 2H). Although quantification did not allow to observe statistically significant effects at lower concentrations, immunohistochemistry showed a slight decrease in MBP staining at 1 µM (Figure 3B). Exposure to BPA and TDCPP resulted in significantly, although not concentration-dependent, reduced myelination at all concentrations tested (Figure 3B,C). It is important to note that these changes in myelination were observed in the absence of changes in NF200 immunostaining. No significant difference was detected in BrainSpheres treated with ibuprofen compared to the DMSO control (Figure 3B,C). In addition, no decrease in the percentage of myelination was observed at eight weeks of differentiation when exposure to test chemicals occurred during the fourth week of differentiation and cultures were allowed to recover for four weeks (data no shown).

Since PLP1 presented a very different distribution than MBP (Figure 4A), it was not possible to use overlapping of its staining with NF staining, as for MBP [57]. Instead, we quantified PLP1 total fluorescence (Figure 4B) using the ImageJ plugin and we performed Western blot (Figure 4C,D). Exposure to non-cytotoxic concentrations of cuprizone (1, 10 and 50 µM), methyl mercury (0.5, 1 and 10 µM), BPA (50 and 100 µM) and TDCPP (0.5, 1, 10 and 50 µM) led to statistically significant reduction of PLP1 in total fluorescence (Figure 4B). Western blot quantification (Figure 4C,D) of PLP1 showed a statistically significant decrease after exposure to methylmercury and TDCPP, however, most of the chemicals (with the exception of ibuprofen) showed a clear down-regulation tendency.

## 3. Discussion

Most human in vitro brain models are not suited to study myelination/demyelination processes, as they do not provide the proper 3D and multicellular CNS microenvironment. Here, we present an iPSC-derived model to study human myelin that includes the main CNS cell phenotypes [61]. BrainSpheres constitute a reproducible human brain model able to recapitulate some of the key events of neurodevelopment, such as proliferation, differentiation of glial cells and neurons, neurite outgrowth, and synaptogenesis [69]. Furthermore, myelination was previously shown to increase progressively with time in culture [61] indicating the usefulness of the model to study this developmental process, in particular since electron microscopy revealed a high level of compaction of myelin sheath around the axons [61]. The gene expression patterns reported in the present study are in line with the ongoing oligodendrocyte maturation and myelination previously observed [61]. In this study, we have established a strategy (myelin quantification, *PLP1* total fluorescence quantification and Western blot) to scrutinize myelin. We propose this strategy to investigate demyelinating diseases and to detect potential NT and DNT compounds. As proof of principle, we have used positive and negative reference chemicals as well as known DNT chemicals to test the capability of the model to identify and quantify in a concentration-response relationship potential myelin disruptor compounds.

To evaluate the use of the BrainSphere model as a myelination test system for DNT evaluation and for disease modelling, we first selected test chemicals with the potential to disrupt oligodendrocytes or myelination (as based on a literature review). Test chemicals, as defined by the ECVAM Biostatistics Task Force in its recommendations for development of alternative toxicological methods, are a set of chemicals of different chemical classes that have been shown to affect the desired endpoint in vivo [96]. In this literature review, in vitro data were also considered due to the possibility that in vivo information would not be available for many chemicals with potential to disrupt oligodendrocytes or myelination.

The myelin quantification assay developed by Kerman and collaborators [57], and a posteriori adapted to BrainSpheres model [61], showed myelination disruption after one week of treatment with cuprizone, but not after the exposure to the negative control compound ibuprofen, emphasizing the ability of this strategy to specifically detect demyelinating chemical agents. Demyelination was also observed after exposure to BPA and TDCPP (Figure 3C) at non-cytotoxic concentrations, reinforcing the previous reports about their potential to induce myelin disruption [72,73,76] and DNT. Methyl mercury chloride significantly decreased myelination only at concentrations reducing cell viability, precluding the identification of methyl mercury as a specific myelin disruptor (Figure 2H). However, this effect may have been missed due to the intervals of concentrations used and the timepoints of assessment.

The effects on myelination were only observed after the exposure to chemicals during the latest window of differentiation of BrainSpheres (Figure 1C and Figure 3C), suggesting that these chemicals target only more mature, deposited myelin and/or that the myelin quantification assay is not sensitive enough to identify very subtly disrupted or malformed myelin—and therefore, may not capture the full effect of the test chemicals on early oligodendrocyte differentiation. For example, decrease in myelination would not be detected in this assessment in case of a simultaneous axonal loss, since the percentage of myelination is relative to the axons in the system. However, this is an advantage of the assay since it is meant to elucidate specific deleterious effects on myelination. We believe that other DNT compounds may also be able to produce effects in early stages of differentiation, so examination of early stage exposure will be important for future DNT evaluation.

Total fluorescence of PLP1 was also a good way to measure the disruption of myelin by the test chemicals (Figure 4B). PLP1 quantification showed similar results as the quantification of myelination. Indeed, all the chemicals (except ibuprofen) reduced PLP1 content. The variation on the sensitivity of the methods used for MBP and PLP1 quantification, might be due to the detection method rather than the mechanism of action of the different compounds. We consider that the combination of all these methods could decrease uncertainties in the screening of potentially toxic compounds, and help compounds safety assessment.

As a myelin study complement, we studied the effect of tested chemicals on oligodendrocytes and astrocytes population, although confocal image analyses were difficult to assess. The morphology and number of O4 positive cells are quite variable between different sections of the same BrainSpheres (Appendix A). Nevertheless, in the confocal images, we observed a decrease of the cell body size of the O4 positive cells (Appendix A) after cuprizone and TDCPP treatment. TDCPP also showed some morphology changes at the higher concentration (50 µM) for S100β marker, showing thicker cell bodies. It is also possible to observe that after the exposure to 1 and 10 µM of methyl mercury and the O4-positive cells phenotype is lost, indicating a possible effect on the viability of the oligodendrocytes present (Appendix A) In addition, the 10 µM concentration of methyl mercury was able to produce some general cytotoxic effects as well.

A few limitations of this study should be stated: The experimental design employed only one cell line. Testing several lines with different genetic backgrounds would add human genetic variability to the test, increasing its robustness. A larger list of chemicals and higher number of experimental replicates will be required to further validate the model. Finally, although this study has been mainly focused on chemical evaluation, we believe that the model could also be used to study diseases related to myelin disruption.

Overall, the results from this study demonstrate that the BrainSphere is a promising model for myelination/demyelination studies using different quantification methods. This study has also provided preliminary data for future performance assessment of the BrainSpheres as a relevant model for DNT testing. Further work is needed to improve the throughput of this assay in BrainSpheres to be able to test more chemicals.

## 4. Materials and Methods

### 4.1. Literature Review to Identify Test Chemicals

A comprehensive literature review was conducted to retrieve studies on exposures to environmental compounds targeting myelination in the CNS during development, with the aim of selecting test chemicals to assess the performance of our endpoints for the myelin assay.

Search strings (Table 1) were combined with the Boolean operator OR and the asterisk (*) was used as a wildcard in the search. The term: “NOT ((“peripheral” [tw]) NOT (“central” [tw])) NOT ((“peripheral nervous system” [mesh]) NOT (“central nervous system” [mesh])) NOT (review [pt]) NOT (“review” [tw]) NOT (“systematic” [sb]) NOT (case reports [pt]) NOT (comment [pt]) NOT (congresses [pt]) AND (English [lang])” was added to the end of the combined search strings to exclude studies exclusively to the peripheral system, exclude review articles, case reports, comments, and congresses, and only include articles in the English language.

#### Study Prioritization and Selection

The PubMed ID (PMID) list of retrieved articles was exported from PubMed and imported into Sciome Workbench for Interactive Computer-Facilitated Text-mining (SWIFT)-Review software (Desktop Version 1.21, Sciome LLC, Durham, NC, USA), which is a freely available software that can be used in literature prioritization. Within SWIFT-Review, the title and abstract of 5% (262 papers) of the retrieved results were manually screened and annotated as “included” or “excluded” based on predetermined inclusion and exclusion criteria (Table 2). These articles were marked as “training items” within the program. In addition to the 5% of articles used to train the machine learning algorithm within SWIFT-Review, another 5% of articles were annotated as “included” but not marked as “training items” to serve as the test-set. The “Prioritize” tool was then used to automatically triage the complete set of articles, and a prioritization score ranging from zero to one was computed for each article by the program, such that the articles similar to the “included” articles marked as “training items” and ranked near the top of the list, and the articles similar to the “excluded” articles are ranked near the bottom. A report of the ranking performance of the algorithm was then generated in SWIFT-Review.

Then, a threshold ranking was selected for abstract and title review within the SWIFT-Review program based on the inclusion and exclusion criteria in Table 2. To generate a list of possible test chemicals (Appendix A), full text review was then conducted on articles included from the title and abstract review process. Test chemicals for assay development were then selected (Figure 1G) based on proposed mechanisms of myelin disruption in the included studies (Appendix A).

### 4.2. Generation of Neural Progenitor Cells from Human Induced Pluripotent Stem Cells

Neural progenitor cells (NPCs) were generated from induced pluripotent stem cells (iPSCs) derived from human CCD-1079Sk fibroblasts (ATCC^®^ CRL-2097) using Epstein–Barr virus-based vectors and embryoid body formation as previously described [97,98] (Figure 2A: NPCs were then plated in 175 cm^2^ Nunc™ EasYFlask™ Cell Culture Flasks (Thermo Fisher, Waltham, MA, USA) coated with poly-l-ornithine (PLO, Sigma-Aldrich, ST. Louis, MO, USA) and laminin from Engelbreth–Holm–Swarm murine sarcoma basement membrane (Sigma-Aldrich) in NPC medium (StemPro^®^ serum-free human neural stem cell culture medium, NSC SFM, GIBCO) containing KnockOut™ D-MEM/F-12 basal medium (GIBCO, Waltham, MA, USA) supplemented with 1% StemPro^®^ NSC SFM supplement (GIBCO), 1% penicillin streptomycin glutamine (GIBCO), 1% Glutamax (GIBCO), 0.01 μg/mL basic fibroblast growth factor (bFGF, GIBCO), and 0.01 μg/mL epidermal growth factor (EGF, GIBCO) [61]). Medium was filtered using a bottle-top filter (Nalgene^®^ Rapid-Flow™ 0.2 µm, 500 mL, Thermo Fisher) prior to the addition of growth factors [61]. NPC cultures were maintained at 37 °C in an atmosphere of 5% CO_2_ and half of the media was changed daily [61]. For their expansion, NPCs were detached mechanically with a cell scraper (2-Posit. Blade 25, Sarstedt, Newton, NC, USA) [61]. NPCs were karyotyped (Appendix A) as well as screened and found to be negative for all strains of mycoplasma (Appendix A).

To generate stocks of NPCs (Passages 15–22), cells were detached mechanically through scraping, placed in STEMdiff™ neural progenitor freezing medium (STEMCELL Technologies, Cambridge, MA, USA), and then transferred to cryopreservation tubes (Nalgene^®^ cryogenic vials, 2 mL, Sigma Aldrich, ST. Louis, MO, USA). Tubes were stored in a cell freezing container (Thermo Scientific™ Mr. Frosty™ Freezing Container, Thermo Fisher, Waltham, MA, USA) containing isopropanol (Sigma-Aldrich) at −80 °C for a minimum of 4 h and then transferred to liquid nitrogen for long-term storage.

### 4.3. Generation of the 3D Human Brain Model (BrainSpheres)

NPCs were thawed and expanded for 2–3 weeks. BrainSpheres were then prepared as previously reported [61]. Briefly, two million NPCs per well were plated in non-coated 6-well plates in 2 mL of NPC medium and incubated at 37 °C, 5% CO_2_ as free-floating cultures under constant gyratory shaking at 88 rpm (Figure 2A). After two days, medium was exchanged to differentiation medium (Neurobasal^®^ electro medium (GIBCO) supplemented with 1% Penicillin Streptomycin Glutamine (GIBCO), 1% Glutamax (GIBCO), 1% B-27^®^ Electrophysiology supplement (GIBCO), 0.02 μg/mL human recombinant glial-cell derived neurotrophic factor (GDNF, Gemini Bio, West Sacramento, CA, USA), and 0.02 μg/mL human recombinant brain-derived neurotrophic factor (BDNF, Gemini Bio, West Sacramento, CA, USA)). Cultures were maintained for eight weeks at 37 °C, 5% CO_2_ under constant gyratory shaking at 88 rpm, and differentiation medium was changed every second day (Figure 1B).

### 4.4. Chemical Exposure

The BrainSpheres (were exposed to the set of five test chemicals: Cuprizone (Sigma Aldrich, #14690), Methylmercury chloride (Sigma Aldrich, #442534), Bisphenol A (BPA; Sigma Aldrich, #239658), Tris(1,3-dichloro-2-propyl) phosphate (TDCPP; Sigma Aldrich, #32951), and Ibuprofen (Sigma Aldrich, ST. Louis, MO, USA, #I4883) at concentrations ranging from 0.1 μM to 100 μM. Stock solutions of the chemicals were prepared in DMSO. BrainSpheres were exposed for seven days in either the fourth or eighth week of differentiation. In the former case, measurements were performed at the end of the treatment week (referred to as 4/4) and after four more post-treatment weeks in absence of any chemical (4/8), and, in the latter case, measurements were performed at the end of the exposure (8/8) (Figure 1C). Medium containing fresh chemicals or vehicle (DMSO) was changed every 48 h. DMSO final concentration in medium was maintained at 0.1% [32]. Experiments were performed in two independent laboratories.

### 4.5. Cell Viability Assessment

Cell viability was assessed using the resazurin assay to select subcytotoxic concentrations with potential to disrupt myelination in BrainSpheres without inducing general cytotoxicity. At the end of the exposure (8/8), BrainSpheres were transferred to 24-well plates in 500 μL of differentiation medium. Then, 5 μL of a 1 mg/mL resazurin sodium salt (Sigma-Aldrich) stock solution in 1× PBS was added to each well, and plates were kept on a gyratory shaker at 88 rpm in a humidified incubator at 37 °C with 5% CO_2_. After incubation for three hours, 100 μL aliquots of sample medium were transferred to a 96-well plate, and the fluorescence of the resazurin metabolite (resorufin) was measured in a multi-well fluorometric plate reader (Cytofluor Multi-well Plate Reader Series 4000, Perspective Biosystem, Framingham, MA, USA) at 530 nm/590 nm (excitation/emission). Cell viability was determined for three replicates by normalization of absorbance measurements from the test samples to the vehicle (DMSO) control samples. Blank subtraction was performed with normal media without cells. Two independent experiments with 3 technical replicates were performed. Replicate are defined as independent wells in a 6-well plate.

### 4.6. Immunocytochemical Staining and Confocal Imaging

BrainSpheres were fixed for 30 min with 4% paraformaldehyde, washed three times with 1× PBS for 10 min each, then incubated in blocking solution (5% normal goat serum (NGS) and 4% 1× Triton in 1× PBS) for two hours on a shaker at room temperature. BrainSpheres were then incubated with primary antibodies (Table 3) diluted in PBS containing 5% NGS and 1% 1× Triton at 4 °C for 48 h.

BrainSpheres were washed with PBS five times for 30 min each and further incubated for one hour with secondary antibodies (goat anti-mouse Alexa Fluor 488 IgG (Invitrogen) or goat anti-rabbit Alexa fluor 568 IgG (Invitrogen, Waltham, MA, USA), 1:250 in PBS containing 5% NGS) on a shaker at room temperature. BrainSpheres were then washed with PBS five times for 30 min each, and nuclei were stained with Hoechst 33342 Trihydrochloride Trihydrate (Invitrogen, 1:10,000 in PBS) for 30 min on a shaker. BrainSpheres were then mounted on glass slides using mounting medium (Immu-Mount, Thermo Scientific, Waltham, MA, USA). Images were taken using a confocal microscope (Zeiss LSM 700 Confocal III and Zeiss LSM 780 GaAsP) and visualized in ZEN Imaging software (Zeiss, Jena, Germany).

### 4.7. Myelin Quantification

BrainSphere myelination was quantified using a protocol adapted from Kerman et al. (2015) [57] for computer-assisted evaluation of myelin formation on ImageJ [57]. Briefly, stacks of confocal images of BrainSpheres that were immunocytochemically stained for NF and MBP were split into single channel images, brightness was adjusted to maximize signal-to-noise ratio, and the images were converted into black and white binary images. Myelination, as defined by the pixels that overlap between the binary single-channel images of oligodendrocytes (MBP staining) and axons (NF staining), was then quantified using the computer-assisted evaluation of myelin formation (CEM) plugin. The output of CEM is percent of myelination, as defined by pixel overlap. At least five images for each condition from at least two independent experiments were analyzed with this protocol. Data are expressed as percent of axons myelinated relative to the vehicle (DMSO) control.

The myelin marker proteolipid protein 1 (PLP1) was evaluated by measuring total fluorescence in ImageJ. To calculate the corrected total cell fluorescence (CTCF) the following formula was used: CTCF = integrated density-area of selected cell X Mean fluorescence of background readings). Data are expressed as percent of CTCF to the vehicle (DMSO) control.

### 4.8. Gene Expression Analysis

Gene expression analysis was conducted using quantitative reverse transcription PCR (qRT-PCR). Briefly, total RNA was extracted from NPCs and BrainSphere samples at two, four, and eight weeks of differentiation, with Invitrogen TRIzol Reagent (Thermo Fisher) according to the TRIzol Reagent User Guide (Invitrogen, Waltham, MA, USA). RNA quantity and purity were determined using NanoDrop 2000c (Thermo Scientific). One microgram of RNA was reverse-transcribed using M-MLV Promega Reverse Transcriptase (Promega, Madison, WI, USA) according to the manufacturer’s recommendations. Gene expression was then evaluated using specific TaqMan^®^ gene expression assays (Table 4; Life Technologies, Carlsbad, CA, USA). Three housekeeping genes were tested (β-actin, 18S and glyceraldehyde 3-phosphate dehydrogenase (GAPDH)) and results were normalized to the most stable (in our results, GAPDH). qRT-PCR was performed on a 7500 Fast Real Time PCR System machine (Applied Biosystems, Waltham, MA, USA). Each sample was analyzed in triplicates. Log_2_ fold change was calculated for biological triplicates using the 2^−ΔΔCT^ method described by Livak and Schmittgen (2001) [99].

### 4.9. Western Blot

One week after exposure to the different compounds, BrainSpheres were washed with PBS and lysed in RIPA Buffer (R0278, Merck, Kenilworth, NJ, USA) with protease and phosphatase inhibitors (78443, Thermo Fisher). Protein content in the lysates was measured with BCA assay (Pierce BCA Protein Assay Kit, 23225, Thermo Scientific). Proteins (40 µg in 30 μL) were separated using 15% acrylamide gel at constant voltage (120 V, 1h30). After transfer to nitrocellulose membranes (BioRad) for 7 min at constant voltage (25 V) under semi-dry condition (Transblot SD semi-dry transfer cell, BioRad, Hercules, CA, USA) using Trans-Blot Turbo Transfer Buffer (BioRad), detection of protein transfer efficiency was performed using MemCode Reversible Protein Stain Kit (Thermo Fisher Scientific) according to the manufacturer’s instructions. Membranes were then blocked with 5% non-fat milk (Sigma, 70166, Balgach, Switzerland) in TBS-Tween 0.1%, during 1h at RT. Then, membranes were incubated with primary antibody (mouse anti-PLP1, 1:500, overnight at 4 °C) diluted in TBS-Tween containing 5% non-fat milk. Membranes were then washed with TBS-Tween 0.1% and labeled with goat anti-mouse peroxidase-conjugated secondary antibody 2h at RT (1/10,000; GE Healthcare). Bands were revealed with a chemiluminescence kit Sirius (BioRad) and scanned with ChemiDoc XRS system (BioRad). Band intensity was quantified with Image J. Data were normalized to total protein levels using MemCode and expressed as percentage of control samples visualized on the same blot. Gels pictures can be found in Appendix A.

### 4.10. Statistical Analyses

Data was normalized to controls and analyzed by using a non-parametric Kruskal–Wallis H test on GraphPad Prism 8 (GraphPad Software, San Diego, CA, USA).

## Figures and Tables

**Figure 1 ijms-22-09473-f001:**
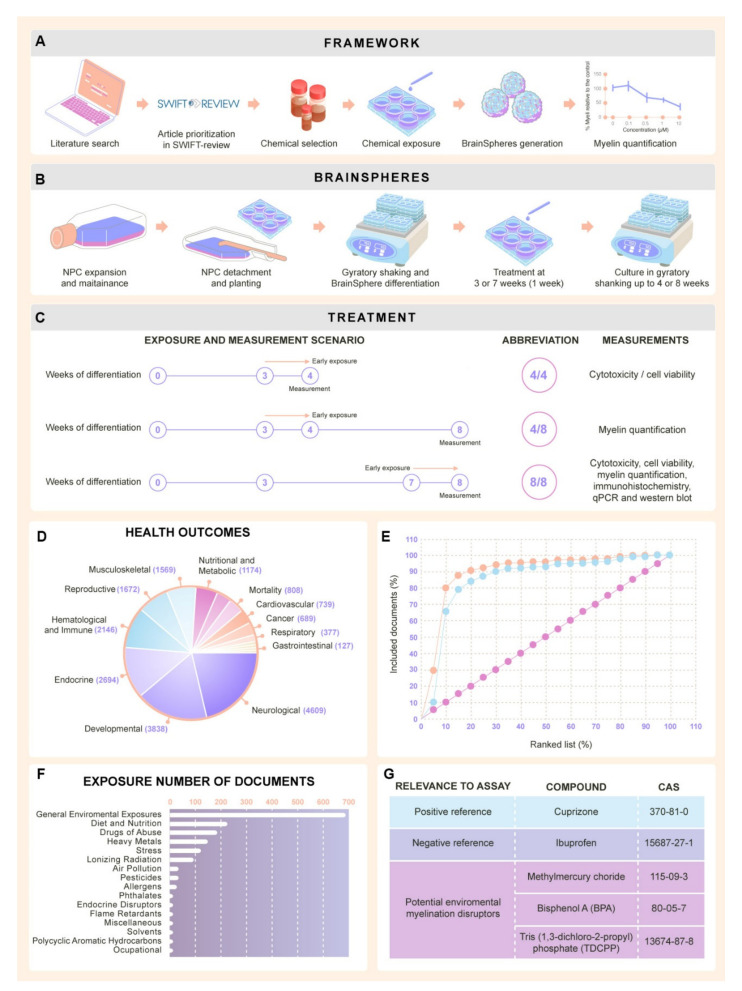
Experimental framework and chemical exposure. (**A**) Experimental framework for myelin quantification. Literature review was performed to select reference compounds and different exposure scenarios were applied. (**B**) BrainSphere preparation [61]. NPCs were expanded, transferred to six-well plates, and kept under constant gyratory shaking for eight weeks to form BrainSpheres. (**C**) Scheme of the three different exposure scenarios used in this study and corresponding abbreviations referenced in text and subsequent figures. (**D**) Pie graph showing the health outcomes repartition of the articles tagged during the SWIFT-Review. (**E**) Ranking performance of the prioritization algorithm in SWIFT-Review based on the publications excluded and included in the training set and included in the test set. The orange line shows the percentage of included documents in the training set as they occur in the ranked list and the blue line shows the percentage of included documents in the test set. The pink line shows the baseline performance if the ranking score had been generated arbitrarily. (**F**) Overview of topics represented in retrieved literature. (**G**) List of chemicals selected for the DNT study. Non-parametric Kruskal–Wallis test was used to determine significant differences between the different conditions.

**Figure 2 ijms-22-09473-f002:**
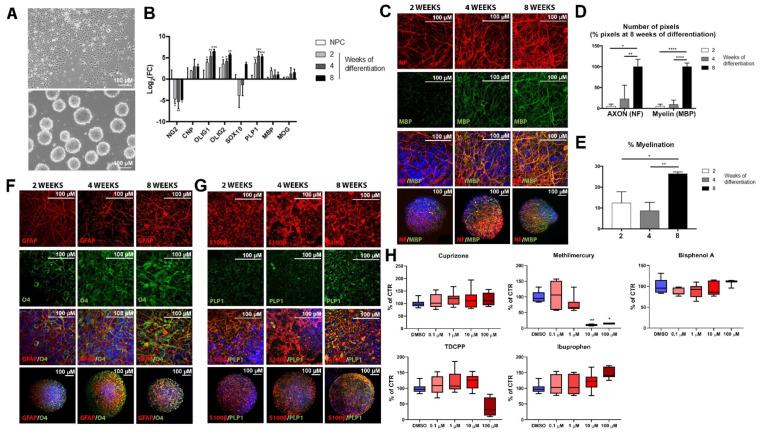
Characterization of glial population in BrainSpheres. (**A**) Representative bright field images of NPCs (upper panel) and BrainSpheres (lower panel). (**B**) Development-dependent oligodendrocyte gene expression in BrainSphere (two, four, eight weeks of differentiation compared to baseline expression in NPCs). Gene expression was normalized to GAPDH. (**C**) Representative confocal images of MBP (myelin marker, green) and NF (axonal marker, red) expression over BrainSpheres differentiation process. (**D**) Number of pixels on confocal images of axonal marker (NF) and myelin marker (MBP) at two, four and eight weeks of BrainSpheres differentiation (mean ± SD, *n* = 10). (**E**) Myelination in BrainSpheres (% of myelinated axons) calculated using CEM plugin at two, four and eight weeks of differentiation. (**F**) Developmental-dependent GFAP (astrocyte marker, red) and O4 (oligodendrocyte marker, green) protein expression. (**G**) Development-dependent S100β (astrocytes marker, red) and PLP1 (myelin marker, green) protein expression. (**H**) Cell viability measured after a seven-day chemical exposure in the eighth week of differentiation (8/8) by resazurin test. Data are expressed as percentage of vehicle (DMSO) control and displayed as mean ± SD. *n* = 6–8 samples coming from two independent experiments. Non-parametric Kruskal–Wallis test was used to determine significant differences between NPCs and BrainSpheres (* *p* ≤ 0.05; ** *p* ≤ 0.01; *** *p* ≤ 0.001; **** *p* ≤ 0.0001). Scale bars = 100 µm.

**Figure 3 ijms-22-09473-f003:**
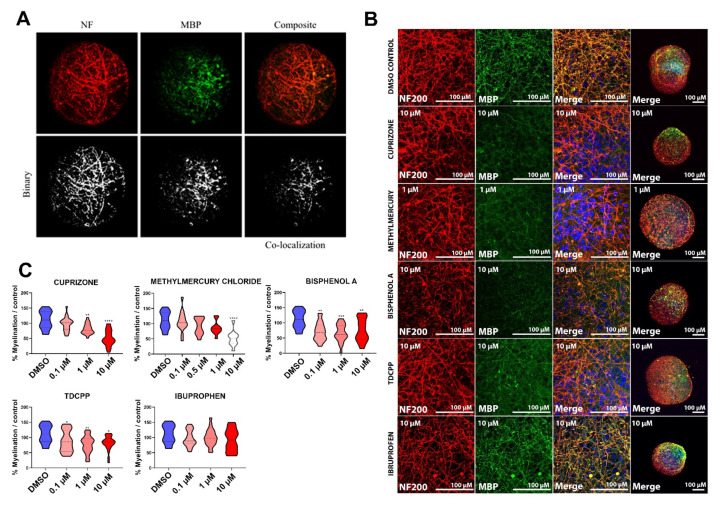
Chemical exposure altered myelin in BrainSpheres. (**A**) Example of binary conversion of confocal images using CEM plugin. (**B**) Confocal images of representative BrainSpheres exposed to the highest non-cytotoxic concentration of each chemical. Treatment, concentration, and antibody used are indicated in each picture. White bars correspond to 50 µm. (**C**) Quantification of BrainSpheres myelination after chemical exposure. Percentage of myelination was quantified as co-localization of MBP and NF immunocytochemical staining in confocal images of BrainSpheres taken at the end of the eighth week of differentiation following a seven-day exposure to test chemicals during eighth (8/8) week. The percentage of myelination was normalized to the mean percentage of myelination in the vehicle (DMSO) control samples. Each bar in the 8/8 experiment represents pooled results from three experiments (first experiment *n* = 5, second experiment *n* = 10, third experiment *n* = 5 total *n* = 20 BrainSpheres). Kruskal–Wallis no-parametric test was used to determine significant differences between control and exposed samples (* *p* ≤ 0.05; ** *p* ≤ 0.01; *** *p* ≤ 0.001; **** *p* ≤ 0.0001). We also performed total quantification of *PLP1*, another myelin marker.

**Figure 4 ijms-22-09473-f004:**
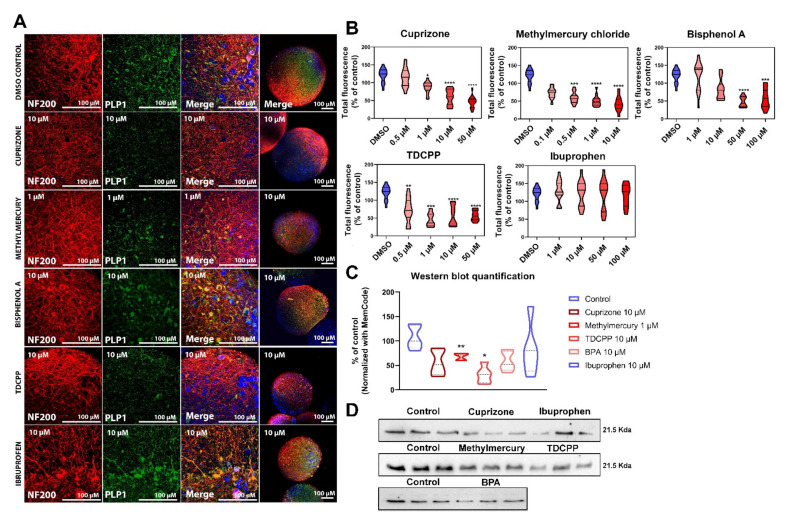
Chemical exposure altered proteolipid protein 1 in BrainSpheres. (**A**) Confocal images of a representative BrainSphere exposed to the highest non-cytotoxic concentration of each chemical. Treatment, concentration, and antibody used are indicated in each picture. White bars correspond to 50 µm. (**B**) PLP1 total fluorescence quantification in percentage of the control. Total fluorescence was measured with ImageJ for PLP1 immunocytochemical staining in confocal images of BrainSpheres taken at the end of the eighth week of differentiation following a seven-day exposure to test chemicals during the eighth (8/8) week. The percentage of PLP1 was normalized to the mean percentage of the vehicle (DMSO) control samples. Each value represents pooled results of two independent experiments with 6 separate BrainSpheres per experiment (*n* = 12). Error bars represent standard deviation. (**C**) Western blot quantification (*n* = 3 per experiment, 2 independent experiments) of PLP1 protein. Samples were normalized with MemCode. (**D**) Representative Western blots Kruskal–Wallis no-parametric test was used to determine significant differences between control and exposed samples (* *p* ≤ 0.05; ** *p* ≤ 0.01; *** *p* ≤ 0.001; **** *p* ≤ 0.0001).

**Table 1 ijms-22-09473-t001:** Search query for PubMed database.

String	Combined with OR		Combined with OR		Combined with OR
1	Embryonic and fetal development, embryonic structures, embryonic, embryo, embryos, embryology, fetal, fetus, pregnancy, gestation, gestational, in utero, prenatal neonatal, neonate, perinatal, postnatal, infant, adolescent, child, fetal brain, human development, developing brain, neurodevelopmental, neurodevelopment	AND	Pharmacological and toxicological phenomena and processes, toxicology, toxicity, toxicity tests, toxicant, toxicants, toxin, toxins, toxic actions, neurotoxicant, neurotoxicants, neurotoxin, neurotoxins, neurotoxins, pharmacology, pharmacologic actions, specialty uses of chemicals, organic chemicals, inorganic chemicals, environment and public health, exposure, exposures, environmental chemical, environmental chemicals, environmental health, hazard, hazards, hazardous, xenobiotics	AND	Myelin, oligodendroglia, myelin sheath, myelinogenesis, myelination, myelin proteins, oligodendrocyte, oligodendrocytes, oligodendrogenesis, white matter, dysmyelination, dysmyelinating, demyelination, demyelinating
2	Developmental neurotoxicant, developmental neurotoxicants, developmental neurotoxin, developmental neurotoxin, developmental neurotoxicity, neurodevelopmental toxicity, neurodevelopmental disorder, prenatal injuries, maternal exposure, teratogen, teratogens, teratogenic				

**Table 2 ijms-22-09473-t002:** Inclusion and exclusion criteria for study selection.

Inclusion Criteria	Exclusion Criteria
DNT is investigated with appropriate exposure scenarios (e.g., prenatal maternal, infant, or childhood exposure (in vivo), exposure during cell proliferation, differentiation, migration, myelination, or synaptogenesis (in vitro))	DNT is not investigated
A single chemical exposure is reported with a clearly identified chemical name or CAS number, or a chemical mixture is reported with human relevance	A chemical exposure is not reported, a mixture of chemicals is reported without human relevance, chemicals are not clearly identified, or exposures are psychosocial (e.g., stress or socioeconomic status), physical (e.g., radiation, particulate matter, or nanomaterials), or intrinsic biological traits (e.g., genetic mutations)
At least two dose or concentration levels are tested or a single dose is tested but was chosen based on previous experience with multiple doses or on human exposure levels	One dose or concentration is tested but was not chosen based on previous experience with multiple doses or on human exposure levels
DNT was evident at doses or concentrations lower than those which cause maternal toxicity (in vivo), general toxicity (in vivo), or cytotoxicity (in vitro)	The relationship between DNT and other forms of toxicity were not described or DNT was only evident at doses that also caused maternal toxicity (in vivo), general toxicity (in vivo), or cytotoxicity (in vitro)
A chemical was characterized as a developmental neurotoxicant using endpoints associated with myelination in the central nervous system (e.g., markers or levels of oligodendrocyte differentiation or MBP gene expression), or a chemical was tested using endpoints associated with myelination during neurodevelopment but found to have no effect	A chemical was not characterized as a developmental neurotoxicant or was characterized using endpoints not associated with myelination, or only with peripheral nervous system myelination
Studies with appropriate negative and solvent controls or control groups	Studies without appropriate negative and solvent controls or control groups

**Table 3 ijms-22-09473-t003:** Primary antibodies.

Name	Abbreviation	Brand	Reference
Neurofilament 200	NF	Sigma	N4142
Myelin basic protein	MBP	BioLegend	808402
Proteolipid protein 1	PLP1	Biorad	MCA839G
Oligodendrocyte marker 4	O4	R&D systems	MAB1326
Glial fibrillary acidic protein	GFAP *	Sigma	G9269
S100 calcium-binding protein B	S100B *	Abcam	ab52642

* Antibody used for results displayed in Appendix A.

**Table 4 ijms-22-09473-t004:** TaqMan^®^ gene expression assays.

Gene Name	Abbreviation	Taqman^®^ Assay ID
Neural-glial antigen 2 (chondroitin sulfate proteoglycan 4)	NG2 (CSPG4)	Hs00361541_g1
Oligodendrocyte transcription factor 1	OLIG1	Hs00744293_s1
Oligodendrocyte transcription factor 2	OLIG2	Hs00300164_s1
SOX-10 transcription factor	SOX10	Hs00366918_m1
Adenomatous polyposis coli	APC	Hs01568269_m1
Proteolipid protein 1	PLP1	Hs00166914_m1
2′,3′-Cyclic-nucleotide 3′-phosphodiesterase	CNP	Hs00263981_m1
Myelin oligodendrocyte glycoprotein	MOG	Hs01555268_m1
Myelin basic protein	MBP	Hs00921945_m1
**Housekeeping Genes**	**Abbreviation**	**Taqman^®^ Assay ID**
β-actin	ACTB	Hs01060665_g1
18S	18S	Hs999999_01
Glyceraldehyde 3-phosphate dehydrogenase	GAPDH	Hs02786624_g1

## Data Availability

Not applicable.

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
