# Peer review of "Human IPSC-Derived Model to Study Myelin Disruption"

_ijms, 2021, doi:10.3390/ijms22179473_

Round 1
Reviewer 1 Report
The authors of this manuscript discuss a relevant research study, focused on the interplay between myelin, environmental risk, and brain development. This is a very important topic which the researchers are well-established to study.
The novel model is of high quality, and overall I found this manuscript of good quality.
I have no major comments. I believe that further discussion on how can we decrease the environmental risks is of high importance, especially given the discussed manuscript. Please elaborate on this topic in the discussion.
Minor comments:
- In the introduction, more articles describing the myelin proteomics' characters can be added (works like of Hauke Werner etc.). Also, when related to white matter pathologies, PMD, autism and recent findings on Williams syndrome and white matter deficits should be added.
- - In the introduction, perhaps a short explanation of the physiological deficits associated with myelin deficits can help explain the importance of myelin in the brain.
- I have a problem with the following statement from introduction: Although co-culturing of neurons and oligodendrocytes exist42-44, most of these cultures 97 are still in a monolayer condition, which does not allow myelination. This is not accurate, and many co-cultures-related studies showed myelination. Please change.
- Did the authors assess normal distribution of data before running the statistical tests?
- Were the experimenters blind? Please specify.
- The figures and their panels should be referred from the main text based on the panels' order in the figure. Figure 2 panels, fpr example, are referred not in order. Please change.
- The text on the figures is in too small fonts. Can't read when printing. Please change.
- No scale bar in Fig. 2A
Author Response
We are thankful that the reviewer likes the manuscript and also for the comments.
- We have added Williams syndrome to the list of white matter pathologies
- We added a phrase indicating that myelin deficits have shown social impairments, motor abnormalities, and poor white matter functioning.
- We have modified the text about co-culturing. We apologize for this mistake.
- We have modified all the graphs to a non-parametric analysis. Data have changed slightly and that is why we have modified also the text. In addition in the case of the myelin quantification, and because we have to re-analyze the raw data we added a 3rd experiment that we haven't had the time to analyze before.
- The experiments were not performed blind
- Figures are now referred in order.
- Text on the figures has been increased as much as possible. We hope now it is easy to see.
- Bars in figure 2 has been added

Reviewer 2 Report
The presented work is relevant, well planned and well executed. It was a pleasure for me to read it.
There are a number of minor flaws: Please arrange the article in accordance with the rules of the journal.
In Fig. 1 it is necessary to indicate the letter designation A, B, C etc. Since they are in the figure description, but not in the figure.
Line 429 (Fig. 3B) replace to Figure.
Author Response
Dear Reviwer,
Thank you for the nice comments.
We have modify the manuscript accordingly to the reviewers:
- We have add letter designation to Figure 1.
- Fig has been changed for Figure in line 429.
Regarding the minor flaws, we will check with the journal.
Best regards,
David Pamies
